# Bleeding into the Abdominal and Ilio-Lumbar Muscles—A Rare Complication in the Course of COVID-19: Analysis of Four Cases and a Literature Review

**DOI:** 10.3390/jcm11164712

**Published:** 2022-08-12

**Authors:** Magdalena Mackiewicz-Milewska, Katarzyna Sakwińska, Małgorzata Cisowska-Adamiak, Iwona Szymkuć-Bukowska, Dorota Ratuszek-Sadowska, Hanna Mackiewicz-Nartowicz

**Affiliations:** 1Department of Rehabilitation, Nicolaus Copernicus University in Toruń, Collegium Medicum in Bydgoszcz, 85-094 Bydgoszcz, Poland; 2Department of Phoniatry and Audiology, Nicolaus Copernicus University in Toruń, Collegium Medicum in Bydgoszcz, 85-168 Bydgoszcz, Poland

**Keywords:** COVID-19 pandemic, critical care, bleeding, anticoagulation

## Abstract

The risk of venous thromboembolic (VTE) complications, mainly in the form of pulmonary embolism (PE) and deep vein thrombosis (DVT), in COVID-19 is well known, necessitating the administration of thrombotic prophylaxis in most patients. With a high risk of VTE complications or their presence, full anticoagulation may be associated with hemorrhagic complications. COVID-19 bleeding is rarely reported. Here, we present four cases of patients with muscle bleeding: two in the iliopsoas muscle, which resulted in death despite the embolization of the bleeding vessel, and two in the oblique and straight abdominal muscles, which were treated conservatively. In the reported cases, the severity of the bleeding coincided with the severity of the course of COVID-19. When observing a sudden drop in hemoglobin (Hb) in a patient with COVID-19, one must always remember the possible complications in the form of muscle bleeding, which can be fatal.

## 1. Introduction

VTE complications, mainly in the form of PE and DVT, in the course of COVID-19 are well known. According to the metanalysis conducted by Tan [1], the incidence of VTE is 14.7%, that of DVT is 11.2%, and that of PE is 7.8%. For this reason, thromboprophylaxis in patients hospitalized due to SARS-CoV-2 infection is a routine procedure, and full anticoagulation therapy is implemented in patients with thromboembolic symptoms or high levels of D-dimer, among others.

It is known that anticoagulation plays an essential role in the treatment of COVID-19, but the optimal dosage and type of preparations to be used are not fully understood and more research is needed [2]. Rizk et al. [3] recommended doses depending on the severity of patient symptoms: a standard dose of anticoagulant for those with moderate COVID-19, and a prophylactic dose for patients with severe COVID-19, due to the higher risk of bleeding. The use of antithrombotic prophylaxis and antithrombotic therapy is associated with possible bleeding complications. Hemorrhagic complications of anticoagulation therapy are known, and most are self-limiting [4]. The risk of bleeding in people without COVID-19 taking prophylactic doses of LMWH (low-molecular weight heparin) is 1.5–2.7%, and in therapeutic doses, it is 1.2–2.1% [5]. However, compared to the high rate of thrombotic complications in COVID-19, hemorrhagic complications are relatively rare; therefore, standard anticoagulation is recommended [6].

Bleeding into soft tissue, especially muscle, in COVID-19 patients, has been relatively rarely reported and may be fatal. From mid-November 2021 to the end of March 2022, i.e., during the operation of the Department for COVID-19 No. 5, temporarily transformed from the Rehabilitation Department, we hospitalized 177 patients, 4 of whom were diagnosed with muscle bleeding, including 2 fatal cases, which constituted 2.3% of patients. Below, we present the four cases of patients with bleeding into soft tissues.

The approval of the local Bioethics Committee of the Nicolaus Copernicus University in Toruń at Collegium Medicum im. Ludwik Rydygier in Bydgoszcz No. KB 405/2022 was obtained.

## 2. Materials

### 2.1. Patient No. 1

A 72-year-old woman was admitted due to pneumonia in the course of a COVID-19 infection. She had a positive PCR test on admission and presented with symptoms of acute respiratory failure, shock requiring Levonor (noradrenalinum) administration, and concomitant hyponatremia (sodium: 110.8 mmol/L). As the interview from the patient was impossible, the interview was supplemented by her daughter. Ten days before admission, the patient was weakened and without fever; the GP clinically diagnosed pneumonia (no PCR test was performed then) and she started a course of Cefuroxime. The patient had the following diseases: treated chronic lymphocytic leukemia with anemia and thrombocytopenia, hypothyroidism, arterial hypertension, chronic hepatitis B, and a condition after stent implantation years ago due to chronic coronary syndrome. Clinically, on admission, the patient was conscious, sleepy, with dyspnea, blood pressure up to 80/50 mmHg, and with an oxygen mask, 15 L/min, at 88% saturation. On the chest CT angiogram, there were no signs of pulmonary embolism; there were inflammatory lesions covering 50% of the lungs on both sides of anopaque glass type. The laboratory results on admission are presented in Table 1 and Table 2. The patient was likely in septic shock. On admission, she received Dexaven 8 mg, Enoxaparine 1 mg/kg every 12 h (due to high D-dimer values and a severe condition), Tazocin 3 × 4.5 g, oxygen through a mask at 10 L/min; no remdesivir was given because of the length of time from the infection. In the following days, verbal contact with the patient improved, the Levonor doses were reduced, and the patient was without dyspnea; the oxygen was kept at 5 L/min, as without oxygen the saturation was 88%. After two days of hospitalization, the D-dimer level increased to 7250 ng/mL and the PCT (procalcitonine) increased to 2.5 ng/mL. A chest CT angiogram was re-performed, without producing evidence of pulmonary embolism. Due to the lack of diagnosis of PE and moderate thrombocytopenia, the dose of Enoxaparine was reduced to 0.5 mg/kg every 12 h, despite the fact that high D-dimer levels above 4200 ng/mL were maintained. The IL 6, CRP, PCT, and WBC also increased. The parameters of the coagulation system are shown in Table 2.

Due to the increase in the inflammatory parameters, despite negative blood and urine cultures, the antibiotic therapy was extended to include Ciprofloxacin. Slow anemization was observed, which was considered secondary to the infection and leukemia. On the third day of hospitalization, there was a sudden drop in blood pressure of 90/60 mmHg, saturation, pallor of the skin, tachycardia above 100 beats/min, deterioration of verbal contact with the patient although conscious, shortness of breath, negative pain, and clinical symptoms of shock. Within one hour, there was a further decrease in oxygen saturation and blood pressure; the patient was intubated and mechanically ventilated. The laboratory studies, on the day of worsening, showed a sudden drop in Hb to 5.8 g/dL, platelets to 50 × 10^3^, and RBC to 1.96 × 10^6^, as presented in Table 1. In an urgent CT of the abdominal cavity, active arterial bleeding into the right iliopsoas muscle (Table 3) from the iliolumbar artery was found. The hematoma was 110 × 92 × 220 mm. There was a puncture of the hematoma into the peritoneum and retroperitoneal space; the blood near the liver and interpetulus in the mesogastrium was up to 20 mm wide, in the lower abdomen up to 40 mm, and around the left kidney 15 mm. The patient qualified for urgent iliopsoas artery embolization, which was performed on the day of bleeding. Enoxaparine was stopped immediately. The next day, there was a large hematoma in the right groin area, and re-embolization of the vessel was observed. Then, four units red blood cells and FFP were transfused. After two days, another CT of the abdominal cavity showed a large hematoma at the visceral surface of the liver, 72 × 38 × 20 mm. There was a hematoma of the extraperitoneal space on the right smaller than before, with dimensions of 115 × 66 × 180 mm. In its center, a circular liquid part of 60 mm was visible, with no noticeable flow. The remainder of the hematoma was clotted. In the following days, another decrease in Hb was observed, and the patient required more blood units. In the following days, the D-dimer level increased to 33,000 ng/mL, but due to the bleeding and thrombocytopenia, LMWH was not administered, and Arixtra was included in a prophylactic dose of 2.5 mg. In the following days, the platelet level ranged from 34 × 10^3^ to 68 × 10^3^. The patient’s condition worsened, and she died seven days after the diagnosis of bleeding and ten days after the diagnosis of COVID-19 due to progressive respiratory failure.

### 2.2. Patient No. 2

A 90-year-old man, admitted due to weakness and a fever of up to 38 °C from the day before admission, was diagnosed with COVID-19 by PCR test. In the interview, the patient disclosed that he was diagnosed with bladder cancer 2 years previously and did not consent to surgery; he also had an abdominal aortic aneurysm. On the day of admission, the patient was in good general condition, walking, with 95% saturation; he did not require oxygen. Remdesivir and Enoxaparine at a prophylactic dose of 0.4 mg s.c were included in the treatment. The thoracic CT angiogram was without signs of pulmonary embolism and the inflammatory lesions typical of COVID-19 in the lungs. The lab tests performed at admission are presented in Table 1. On the ninth day of hospitalization, the patient presented weakness and a decrease in saturation to 89%, and he required oxygen therapy. A CT of the chest was performed, showing densities on both sides of a matte glass type and less numerous parenchymal lesions scattered in the lungs, covering about 40% of the entire pulmonary parenchyma, a progression compared to the previous examination. In the thoracic CT angiogram, there was an abdominal aortic aneurysm, with dimensions of 76 × 62 mm, with an inhomogeneously dense eccentric thrombus up to 36 mm thick on the anterior wall, which could suggest the possibility of a near rupture of the AAA (abdominal aortic aneurysm). There was no evidence of a pulmonary embolism. Due to the persistence of high D-dimer values (4500–8500 ng/mL) and a thrombus in the abdominal aortic artery, the dose of Enoxaparine was increased to 1mg/kg every 12 h. After another two days, there was a further decrease in saturation; the oxygen therapy was escalated, initially with high flow, then noninvasive ventilation with a mask in PCV mode; on the following days, the patient was intubated and mechanically ventilated from the sixteenth day of hospitalization. On the twentieth day of hospitalization, there was a decrease in Hb to 7.7 g/dL, RBC to 2.57 × 10^6^, and PLT to 104 × 10^3^.The selected laboratory tests at the admission and during bleeding are presented in Table 1 and Table 2. Imaging diagnostics were undertaken (an abdominal ultrasound, in which no free fluid in the abdominal cavity was visualized), as well as a surgical consultation and an abdominal CT scan, in which the presence of blood in the lumen of the large intestine could not be ruled out. The patient received two units of RBCs (packed red blood cells), followed by gastroscopy, which showed no evidence of fresh or previous bleeding. In the following days, there was a further decrease in Hb; imaging diagnostics were resumed (four-phase CT of the chest, abdomen, pelvis, and thighs), and a hematoma in the right iliac muscle (Table 3) was found, as well as retroperitoneal bleeding. Moreover, there was a progression of the lung parenchyma involvement to 70%. The patient was consulted surgically and then with an interventional radiologist, after which the patient was qualified for a bleeding vessel embolization procedure. The embolization of the right hip and L4 lumbar artery was performed without complications. The anticoagulation treatment was discontinued. However, the patient’s condition gradually deteriorated and increasing multiorgan failure was observed. Due to the suspicion of an embolism component, the chest CT angiogram scan was performed again, in which no embolism was shown; progression of lesions typical for COVID-19 to 80–90% of the lung parenchyma was found. After thirty days of hospitalization and four days after the diagnosis of bleeding, the patient died due to progressive respiratory failure.

### 2.3. Patient No. 3

A 49-year-old woman was admitted to the COVID-19 department due to a SARS-CoV-2 infection confirmed by PCR test. She came to the local hospital because of the swelling and redness of her left foot that had been increasing for a week. In the days preceding the admission, she experienced significant cooling in the lower limbs. She denied other diseases and chronic medication intake. A CT angiogram of the lower extremities was performed, and a narrowing of the lumen of the right common iliac artery by about 30% was found (by soft-tissue plaques/thrombus), along with a stenosis of the right popliteal artery by about 40% and of the left by about 20% strictures. In the CT examination of the chest, several foci of a matte glass type with reticulated densities were found in the lower parts of the right lung, which may correspond to changes in the course of SARS-CoV-2 infection; the lesions accounted for <2% of the lung parenchyma. The results of the lab tests when admitted were without deviations, except for the increased levels of CRP and D-dimer; the results are presented in Table 1. Enoxaparine was included in the treatment after consultation with a vascular surgeon at therapeutic doses of 1 mg/kg every 12 h. On admission to the department, the patient showed no dyspnea, no cough, and no fever, with a swollen left foot and blue toes. During the stay in the ward, symptomatic treatment was introduced; apart from LMWH, Ceftriaxone was administered, and oxygen therapy was not required. Physiotherapy was carried out. Due to the persistence of the swelling of the left lower limb, Doppler ultrasound examination of the lower limbs was performed, and DVT was excluded. Due to the occurrence of sudden pain in the area of the left hypochondrium, on the sixth day of hospitalization, an ultrasound examination of the abdomen was performed, and in the left abdomen in the rectus abdominis muscle two hematomas 17 × 23 × 34 mm and 18 × 23 × 26 mm in size were found, which were confirmed by a CT of the abdomen (Table 3). In the lab studies (Table 1), a decrease in Hb was observed. The levels of the coagulation system parameters during bleeding are presented in Table 2. The LMWH dose was reduced to a prophylactic dose, and in the following days a reduction in the size of the hematoma and the withdrawal of pain were observed. After reducing the dose of LMWH, there was a transient increase in D-dimer level to 6928 ng/mL and an increase in the PLT to 477 × 10^3^. As a result of the applied treatment, the condition of the left foot was significantly improved, the swelling and redness were withdrawn, and only superficial necrotic scabs on fingers II–IV remained. The stay was complicated by Clostridium difficile infection. After treatment and isolation were completed, the patient was discharged home in good general condition.

### 2.4. Patient No. 4

A 76-year-old woman was transferred to the COVID-19 department from the Endocrinology and Diabetology Clinic because of a positive PCR test for SARS-CoV-2. The patient reported the presence of dyspnea and cough. According to the medical history, the patient had COPD (Chronic Obstructive Pulmonary Disease), emphysema, chronic circulatory failure, chronic atrial fibrillation, a condition after surgical treatment for cervical cancer in 2012 with radiotherapy, and atherosclerosis of the lower limbs. On admission, the physical examination revealed a good general condition, reduced oxygen saturation to 90%, present leg edema, and auscultation over the lung fields of crackles and rales. The chest CT performed on the day of admission did not show lesions typical for COVID-19 infection, but a confluent consolidation in the left lung with minor calcifications was visible and noted for further control. The laboratory test results on admission are presented in Table 1 and Table 2; moreover, there was hypokalemia, increased concentration of NtproBNP (N-terminal (NT)-pro hormone B-type natriuretic peptide)**,** and lymphocytopenia. During hospitalization, the patient received treatment with remdesivir, IV steroid therapy, antibiotic therapy with Ceftriaxsone IV and Enoxaparine at a therapeutic dose of 1 mg/kg every 12 h (because of AF—atrial fibrillation) and oxygen therapy. Due to the intensification of pain in the abdominal cavity, a CT scan of the abdominal cavity was performed, in which the features of median arcuate ligament syndrome of the diaphragm, RIIA (right internal iliac artery) and LEIA (left external iliac artery) obstruction, and right adrenal adenoma were visualized. She was consulted by a vascular surgeon for further inspection. Due to the increase in inflammatory parameters with a concomitant drop in blood pressure, she required the administration of noradrenaline, Levonor with a low flow, and blood and urine cultures were collected twice. On their basis, staphylococcal sepsis was diagnosed; Linezolid was introduced as a targeted therapy, the inflammatory parameters decreased, and the pressure was stabilized. Then, Levonor was discontinued. On the eleventh day of hospitalization, due to the appearance of acute point pain in the lower abdomen on the left side, increasing with coughing, and progressive anemization (Table 1), an abdominal CT scan was performed; a small hematoma of 15 mm and thickening of the left external oblique muscle were observed (Table 3). After the examination was performed, she was consulted surgically. Due to the appearance of a focus of small hard resistance in the pain relapse, an ultrasound of the abdominal cavity was performed on the next day; the hematoma was enlarged to 40 × 20 mm, and she was surgically consulted again. Due to the reported pain in the left lower limb, including the shin, a Doppler ultrasound was performed, without producing evidence of thrombosis. Due to the high risk of thromboembolic complications (AF), the treatment with Enopxaparine was continued, but the dose was reduced from 1 mg/kg every 12 h to 0.75 mg/kg every 12 h. The abdominal pain subsided, and the hematoma decreased. After the isolation period, the patient was discharged to the Department of Cardiology for further treatment.

## 3. Discussion

We have reported four cases of bleeding in the course of COVID-19 that show that major bleeding, especially into the retroperitoneal space, was a fatal complication, possibly due to the anticoagulation used due to thromboembolic complications or a high risk of their occurrence, which mainly affected people with multiple diseases (three out of four patients). It cannot be ruled out that the SARS-CoV-2 virus itself increased the risk of bleeding, although it is true that all bleeding patients were on the full therapeutic dose of LMWH or an intermediate dose. Further research in this direction is required. Al-Samkari et al. [7] examined 400 patients diagnosed with COVID-19 and found VTE in 4.8% of patients, 7.6% of whom were critically ill patients. Hemorrhagic complications occurred in 4.8% of patients, including 5.6% of critically ill patients. Most of the bleeding was related to the gastrointestinal tract (no bleeding into the iliopsoas or abdominal muscles was observed in any of the patients). The authors of the study reported that the risk of VTE and bleeding was comparable in patients with COVID-19, but further randomized studies are needed to determine the optimal dose of thromboprophylaxis in patients with COVID-19. In a review of the literature, we found some case reports of IPH (iliopsoas hematoma) [4,8,9,10,11,12,13].

Riu et al. [13] observed soft tissue bleeding in 21 patients out of 1750, which accounted for 1.95% of patients hospitalized for COVID-19. The most frequently described soft tissue bleeding sites were the iliopsoas in 11 patients and the rectus abdominal muscle in 3 patients. Of the 21 patients, 20 received an anticoagulant therapeutic dose, initially prophylactic. Moreover, Vergori et al. [9] described a large group of seven cases of iliopsoas bleeding in 925 patients hospitalized with COVID-19. Abate et al. [10] observed 10 cases of spontaneous muscle bleeding (SMH) in 475 hospitalized COVID-19 patients. The mortality rate of SMH in COVID-19 patients was 32.4%. The sites of bleeding were the iliopsoas, vastus intermedius, gluteus, sternocleidomastoid, and pectoralis major muscles. Prevalence of SMH in the study group in the Abate research was comparable to our group—2.1%, and 2.3%, respectively. The most common clinical symptoms reported by the authors were symptoms of sudden anemia, general symptoms of hypovolemia, and neurological symptoms due to ipsilateral pressure on the lower limbs [4,9,11]. In two cases of iliopsoas muscle bleeding with retroperitoneal bleeding, a sudden drop in Hb was observed, and in one patient, there were also symptoms of hypovolemia with a sudden drop in blood pressure requiring the administration of pressor amines. However, no neurological disorders were observed, which may have resulted from the inability to determine this due to the serious condition of the patients. Patient No. 2, at the time of bleeding, was intubated, sedated, and mechanically ventilated, and Patient No. 1, at the time of massive bleeding, was intubated, and sedation and mechanical ventilation were initiated. Wada et al. [14], as a result of the conducted research, stated that platelet activation may be important in the worsening progress of COVID-19.

In terms of bleeding into the abdominal muscles and the rectus and oblique muscles, the described cases were definitely less critical. Indeed, anemization of the patients was observed, but it was not as severe as that for the abovementioned patients; moreover, the hematomas were self-limited. Nevertheless, the main symptom was abdominal pain, and bleeding into the muscles should always be taken into account when diagnosing COVID-19 patients with abdominal pain. In the literature review, we found two reports of bleeding events in the abdominal muscles in a patient in the course of COVID-19 upon anticoagulant treatment [15,16].

## 4. Limitations of the Study

The limitation of the study was the lack of a control group, i.e., patients hospitalized in the internal medicine or infectious diseases ward without COVID-19 infection.

## 5. Conclusions

Muscle bleeding is rare but can be a serious, even fatal, complication of COVID-19 in patients receiving LMWH anticoagulant therapy. In the cases we have described, the severity of bleeding coincided with the severity of the course of COVID-19. When observing a sudden drop in Hb in a patient with COVID-19, one must always remember the possible complications in the form of muscle bleeding, which can be fatal.

## Figures and Tables

**Table 1 jcm-11-04712-t001:** Basic laboratory parameters on admission and after onset of the bleeding.

Pat. No.	Age	Hb 1 (g/dL)	Hb 2	RBC 1 (10^6^)	RBC 2	PLT 1 (10^3^)	PLT 2	CRP 1 (mg/L)	CRP 2
1	72	11.6	5.8	4.19	1.96	106	50	114	128
2	90	12.2	7.7	3.76	2.57	106	104	25.8	15.3
3	49	14.4	11.6	4.6	3.7	205	270	85.4	2.36
4	76	13.6	9.7	3.8	3.02	213	141	13.6	9.18

Legend: Pat.—patient, Hb 1—hemoglobin level on admission, Hb 2—level during bleeding (references values: 11.2–15.7 g/dL), RBC 1—red blood cell level on admission**,** RBC 2—level during bleeding (references values: 3.93–5.22 × 10^6^), PLT1—level on admission, PLT 2—level during bleeding (references values: 132–370 × 10^3^), CRP 1—C reactive protein level on admission, CRP 2—level during bleeding (references value: <5 mg/L).

**Table 2 jcm-11-04712-t002:** Coagulation laboratory parameters on admission and onset of the bleeding.

Pat No	D-Dimer 1 (ng/mL)	D-Dimer 2	APTT 1 (s)	APTT 2	Fibrinogen 1 (mg/dL)	Fibrinogen 2	PT 1 (s)	PT 2
1	4704	4200	29.1	30.2	285	166	13.1	16.2
2	12,880	7200	38.1	55.2	167	169	13.6	12.4
3	1238	1635	x	35.2	x	419	x	11.4
4	673	1275	29.7	29.3	673	665	16.2	12.1

Legend: Pat.—patient, D-dimer 1—level on admission, D-dimer 2—level during bleeding (reference value: <500 ng/mL), APTT 1 activated partial thromboplastin time on admission, APTT 2 activated partial thromboplastin time during bleeding (reference values:24.0–36.0 s), Fibrinogen 1 level on admission, Fibrinogen 2 level during bleeding (reference values: 200–399 mg/dL), PT1 prothrombin time on admission, PT 2 prothrombin time during bleeding(reference values: 10.2–12.9 s), x—the laboratory test was not performed.

**Table 3 jcm-11-04712-t003:** Characteristics of patients (anticoagulation during bleeding), site of bleeding, treatment of bleeding, lung involvement, and further fate of patients).

Patient No.	Bleeding Area	Anti-Coagulation LMWH (Enoxaparine)	Treatment	Lung Involvement (%)	Further Fate	Blood Transfusion (Unit)
1	Right iliopsoas muscle	2 × 0.5 mg/kg	Embolization	40	Death	4
2	Right iliac muscle	2 × 1 mg/kg	Embolization	70	Death	2
3	Rectus abdominis left muscle	2 × 1 mg/kg	Conservative	<2	Home	0
4	Left oblique abdominal muscle	2 × 1 mg/kg	Conservative	0	Cardiology clinic	0

## Data Availability

The data presented in this study are available on request from the corresponding author. The data are not publicly available due to privacy restrictions.

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
