# Peer review of "Bleeding into the Abdominal and Ilio-Lumbar Muscles—A Rare Complication in the Course of COVID-19: Analysis of Four Cases and a Literature Review"

_jcm, 2022, doi:10.3390/jcm11164712_

Round 1
Reviewer 1 Report
The authors describe a case series of 4 patients admitted with COVID-19 who developed muscle bleeding, 2 of whom subsequently died.
Overall the cases are poorly described, and no information is given, for example, of relevant coagulation assays (PT/PTT/fibrinogen), or whether patients were on dialysis. There are a host of other issues, e.g., abbreviations such as PCT, TAB, RIAA and LEIAW are introduced into the main text without explanation, and dosing of enoxaparin is normally expressed not as 0.4 or 0.8 ml of “clexane” but as mg/kg (0.5mg/kg, 1mg/kg) and normally given every 12 hours, not once daily so peaks are not as high.
In general, the English language and style is also so poor it detracts from the message. In addition, scattered throughout the text, is abundant evidence of a lack of attention to detail. For example, the phrase is SARS-CoV-2, not SarsCoV 2, Sars Cov2, SARS CoV-2 or other variants I may have missed. Similarly, it is COVID-19, not COVID 19, COVOD 19, or Covid 19.
Author Response
Dear Reviewer
Thank You very much for Your profound and detailed analysis of our study. We are pleased to implement the changes You recommended.
- Overall the cases are poorly described, and no information is given, for example, of relevant coagulation assays (PT/PTT/fibrinogen), or whether patients were on dialysis.
Ad 1. In the text, we have included Table 2, which contains the parameters of the coagulation system of the described patients.
Table 2. Coagulation laboratory parameters on admission and onset of the bleeding.
Pat No |
APTT 1(s) |
APTT 2 |
Fibrinogen 1 (mg/dl) |
Fibrinogen 2 |
PT 1 (s) |
PT2 |
1 |
29.1 |
30.2 |
285 |
166 |
13.1 |
16.2 |
2 |
38.1 |
55.2 |
167 |
169 |
13.6 |
12.4 |
3 |
x |
35.2 |
x |
419 |
x |
11.4 |
4 |
29.7 |
29.3 |
673 |
665 |
16.2 |
12.1 |
Legend: Pat.-patient; APTT 1 activated partial thromboplastin time on admission ; APTT 1 activated partial thromboplastin time during bleeding( references. 24.0-36.0 s); Fibrinogen 1 level on admission , Fibrinogen 2 level during bleeding ( references 200-399); PT1 prothrombin time on admission , PT 2 prothrombin time during bleeding( references. 10.2-12.9); x- the laboratory test were not marked.
- There are a host of other issues, e.g., abbreviations such as PCT, TAB, RIAA and LEIAW are introduced into the main text without explanation, the abbreviation was explained .
Ad. 2 We are very sorry, but we used the Polish abbreviation TAB instead of AAA for Abdominal Aortic Aneurysm. It has been corrected. Other abbreviation has been explained.
No patient was dialysed.
- Dosing of enoxaparin is normally expressed not as 0.4 or 0.8 ml of “clexane” but as mg/kg (0.5mg/kg, 1mg/kg) and normally given every 12 hours, not once daily so peaks are not as high.
Ad. 3 We have corrected the Enoxaparine dosing record; everywhere the Trade Name was changed to chemical from Clexane to Enoxaparine and doses in mg / kg were given, except for the prophylactic dose, which is routinely administered once a day at a dose of 0.4 mg s.c.
- In general, the English language and style is also so poor it detracts from the message. In addition, scattered throughout the text, is abundant evidence of a lack of attention to detail. For example, the phrase is SARS-CoV-2, not SarsCoV 2, Sars Cov2, SARS CoV-2 or other variants I may have missed. Similarly, it is COVID-19, not COVID 19, COVOD 19, or Covid 19.
Ad 4. Thank you for pointing out the heterogeneity of the spelling of COVID-19 and SARS-CoV-2. The errors have been corrected.
Text editing was done by the professional english editor MDPI.

Reviewer 2 Report
The authors have presented a paper titled Bleeding into the abdominal and ilio-lumbar muscles - a rare complication in the course of COVID 19 - analysis of 4 cases and literature review. However, some issues regarding the manuscript are listed below.
1. Please be consistent with the disease name, whether COVID 19 or Covid19
2. There were many misspelled words and inappropriate use of abbreviations—however, it is hard to point out in detail since there is no line marking in the manuscript.
3. Table I title was not added correctly, and the table legend is confusing.
4. There was an inconsistent style of writing.
5. The case presentation is unclear, And the patient history was hard to follow.
In summary, this paper needs some improvement in terms of writing and improvement on its content.
Author Response
Dear Reviewer
Thank You very much for Your profound and detailed analysis of our study. We are pleased to implement the changes You recommended.
- Please be consistent with the disease name, whether COVID 19 or Covid19
Ad1. The change has been made.
- There were many misspelled words and inappropriate use of abbreviations—however, it is hard to point out in detail since there is no line marking in the manuscript.
Ad 2. The abbreviations has been explained. The spelling errors has been corrected.
- Table I title was not added correctly, and the table legend is confusing.
Ad.3 The proper name has been added. We have changed the legend hoping that bit would be clear.
Legend: Pat.-patient; Hb1 - hemoglobin level on admission, Hb2- level during bleeding( references 11.2-15.7 g/dl); RBC1 Red Blood Cell level on admission, RBC2- level during bleeding ( references 3.93-5.22 x106); PLT1 –Plate Counts level on admission, PLT2- level during bleeding (references 132-370 x103); D-dimer1- level on admission, D-dimer2- level during bleeding ( references <500 ng/ml); CRP1- C reactive protein level on admission, CRP2- level during bleeding (references <5mg/l).
- There was an inconsistent style of writing. The case presentation is unclear, And the patient history was hard to follow.
Ad 4 and 5 Text editing has been done by the professional english editor MDPI.

Round 2
Reviewer 2 Report
The authors have improved their manuscript. This paper is beneficial to enhance the awareness of medical personnel who treat COVID19 patients. However, these were some minor issues that need to address as follows;
Line 66: what is RR 80/50 mmHg? is it blood pressure? avoid abbreviations without explanation.
Line 113: Please use "seven" instead of "7" also in line 162, 186,
Line 142: Please use the unit --> 38 0C
Author Response
Thank you very much for valuable comments. I have made corrections to the currently submitted version of the manuscript as suggested, the linguistic errors have been removed.
ad.1 Line 66 blood pressure has been added instead RR
ad.2 linguistic correction has been madead.
3 the fever was given in degrees 38 °C
